# Automated Counting of Tobacco Plants Using Multispectral UAV Data

**Hong Lin** [1,†] , **Zhuqun Chen** [1,†], **Zhenping Qiang** [1,*], **Su-Kit Tang** [2], **Lin Liu** [3] and **Giovanni Pau** [4,5]

1 College of Big Data and Intelligent Engineering, Southwest Forestry University, Kunming 650224, China; linh1226@swfu.edu.cn (H.L.); chenzhuqun2023@163.com (Z.C.)
2 Faculty of Applied Sciences, Macao Polytechnic University, Macao SAR 999078, China; sktang@mpu.edu.mo
3 College of Tobacco Science, Yunnan Agricultural University, Kunming 650500, China; liulin6032@163.com
4 Department of Computer Science, University of Bologna, 40126 Bologna, Italy; giovanni.pau@unibo.it
5 Samueli Computer Science Department, University of California, Los Angeles, CA 90095, USA
* Correspondence: qzp@swfu.edu.cn
† These authors contributed equally to this work.

**Abstract:** Plant counting is an important part in precision agriculture (PA). The Unmanned Aerial Vehicle (UAV) becomes popular in agriculture because it can capture data with higher spatiotemporal resolution. When it is equipped with multispectral sensors, more meaningful multispectral data is obtained for plants' analysis. After tobacco seedlings are raised, they are transplanted into the field. The counting of tobacco plant stands in the field is important for monitoring the transplant survival rate, growth situation, and yield estimation. In this work, we adopt the object detection (OD) method of deep learning to automatically count the plants with multispectral images. For utilizing the advanced YOLOv8 network, we modified the architecture of the network to adapt to the different band combinations and conducted extensive data pre-processing work. The Red + Green + NIR combination obtains the best detection results, which reveal that using a specific band or band combinations can obtain better results than using the traditional RGB images. For making our method more practical, we designed an algorithm that can handling the image of a whole plot, which is required to be watched. The counting accuracy is as high as 99.53%. The UAV, multispectral data combined with the powerful deep learning methods show promising prospective in PA.

**Keywords:** plant counting; tobacco; UAV; multispectral data; objection detection; YOLOv8

## 1. Introduction

Precision agriculture (PA) [1,2] involves the observation, measurement, and response to inter- and intra-field variability in crops, etc. Precision farming may play an important role in agricultural innovation [3]. PA offers a data-driven and technology-enabled approach to optimize farming practices, increase sustainability, and enhance productivity in the face of field variability. It empowers farmers to make precise and informed decisions, leading to improved crop yields.

Plant counting plays an important role in PA. It traverses through almost every critical stage in agricultural production, spreading from seed breeding, germination, cultivation, fertilization, pollination, to yield estimation and harvesting [4]. There are many key reasons to signify the significance of plant counting. Accurate plant counting helps estimate crop yields, which is crucial for agricultural planning, resource allocation, and market forecasting. It enables farmers to make informed decisions regarding harvesting, storage, and the marketing of their produce. Yield estimation should start from (1) crop management: Plant counting provides essential information for effective crop management. Knowing before harvesting how many plants have emerged and how they are growing is key in optimizing labor and an efficient use of resources [5]. By knowing the population density of plants, farmers can optimize irrigation, fertilization, and pest control practices tailored to

specific crop requirements. (2) Plant spacing and thinning: Proper plant spacing is vital for optimal growth and resource utilization. Plant counting helps determine if plants are evenly spaced, allowing for efficient light penetration, air circulation, and nutrient absorption. It also aids in identifying overcrowded areas, facilitating thinning or replanting actions to achieve the desired plant density. (3) Research and experimentation: Plant counting is crucial in scientific research and experimental studies. It helps researchers assess the effects of different treatments, interventions, or genetic modifications on plant growth, development, and productivity. Accurate plant counting enables reliable and meaningful data analysis, leading to valuable insights and advancements in plant science. (4) Disease and pest monitoring: Plant counting can assist in the early detection and monitoring of plant diseases and pests. Plant counting has a wide range of applications in agriculture, such as crop management, yield estimation, disease and pest monitoring, etc., [6]. By regularly counting plants, farmers or researchers can identify and track the spread of diseases or infestations. Timely intervention measures can be implemented to prevent further damage and minimize crop losses. (5) Plant breeding and genetics: Plant counting is essential in breeding programs and genetic studies. It aids in evaluating plant traits, such as flowering time, fruit set, or seed production, and helps select superior individuals for further breeding. Accurate counting enables breeders to make informed decisions in developing new varieties with desired characteristics [7].

The traditional counting is conducted by humans, which is a labor-intensive, time-consuming, and expensive work. The current technologies, such as the Internet of Things (IoT) and artificial intelligence (AI), as well as their applications, must be integrated into the agricultural sector to ensure long-term agricultural productivity [8]. These technologies have the potential to improve global food security by reducing crop output gaps, decreasing food waste, and minimizing resource use inefficiencies [8]. Computer vision provides a real-time, non-destructive, and indirect way of horticultural crop yield estimation [9]. Therefore, we choose to combine deep learning for statistical research on plants. Along with the booming development of artificial intelligence and computer vision, it has become more feasible to monitor the crops by using imagery. For monitoring the stand of crops, the best perspective image is the orthomosaic image, which can be obtained via aerial photography. The unmanned aerial vehicles (UAV), commonly known as drones, is a feasible solution for capturing the images. In the field of plant protection, drones are attracting increasing attention due to their versatility and applicability in a variety of environmental and working conditions [10]. The benefits of using UAV instead of using satellite data is that the data captured via UAV has better spacial resolution, as well as the temporal resolution is more flexible [11]. By equipping it with multispectral sensors, the drone can capture the multispectral imagery, which includes richer information for crop growth observation and estimation [12,13]. Furthermore, drones have been used for agricultural crop research. For example, this study aimed to evaluate the response of coffee seedlings transplanted to areas subjected to deep liming in comparison to conventional (surface) liming, using vegetation indices (VIs) generated by multispectral images acquired using UAVs [14].

In existing works [5,15] of plant counting, the most frequently used image is the three visible light (VIS) bands: blue band (400 to 500 nm), green band (500 to 600 nm), and red band (600 to 700 nm), which cover the range of wavelengths that are visible to the human eye. Except for the visible light bands, multispectral bands can be captured via UAV if it is equipped with multispectral sensors. The essence of multispectral data lies in the reflection of light at various discrete wavelengths, which reflects the reflectance of a point for a particular wavelength. Therefore, the multispectral data can be regarded as images. Deep learning techniques for image processing can also be applied to the multispectral images [16].

Some spectral bands are significant for agriculture and provide valuable information for crop monitoring and analysis [12]. For example, the near-infrared (NIR) band (700 to 1300 nm) provides insights into plant health, photosynthetic activity, and water content.

It is particularly useful for monitoring vegetation vigor, detecting stress conditions, and estimating biomass in crops [17]. The red-edge (RE) band (700 to 750 nm) captures subtle changes in chlorophyll absorption and can indicate growth stages, nutrient status, and stress conditions in crops. It is useful for differentiating between healthy and stressed vegetation [18]. Hence, for the specific purpose of plant counting, we propose to take advantages of using more relevant bands in the multispectral range besides the RBG.

In recent years, along with the blooming of advanced technologies, such as artificial intelligence, computer vision, machine learning, deep learning, etc., they have been greatly introduced into the agriculture industry [19]. Many traditional tasks in agriculture, such as plant counting, plant disease identification, etc., have achieved excellent performance [20,21]. Deep learning is a kind of data-driven method which generally requires large-scale data to train the network. By using UAV, the data collection work is easier and faster. Combined with computer vision, the automated plant counting becomes promising by using the deep learning method.

Tobacco is a kind of highly valuable crop due to its economic significance in the global market. Tobacco cultivation and the tobacco industry contribute significantly to the economies of many countries. It provides income and employment opportunities for farmers, laborers, and workers involved in various stages of production, processing, and distribution. The global tobacco market is substantial, meaning that tobacco products continue to be in demand worldwide. In addition, tobacco is an important commodity for export earnings and value-added products [22].

The cultivation of tobacco occurs annually. The plant is germinated in cold frames or hotbeds firstly and then transplanted to the field. The time from transplanting to harvest depends on the breeds, generally around 60 to 90 days, but in the range of 45 to 120 days [22]. During cultivation in the field, the UAVs can be used for monitoring the number of stands after the transplanting, as well as the number of plants during the growth in field. The yield of tobacco largely depends on the number of viable tobacco plants because the leaves are what is harvested from the tobacco plant. Hence, tobacco plant counting is also relevant for the yield estimation.

In this work, we propose to utilize the images captured via UAV for automatic tobacco plant counting. The images not only include the visible images, but also include the multispectral images. Our contributions of this work can be expressed as follows: (1) we proposed to use the multispectral images for plant counting; (2) we created a dedicated multispectral tobacco plant dataset; (3) Because our input data has three different channels, we have modified the architecture of YOLOv8 and designed a post-processing algorithm to count the stands of a big field.

## 2. Review of Related Work

### 2.1. Plant Counting

In recent years, the UAV is more frequently used for plant counting because the captured data has higher spatial and temporal resolution [23]. João Valente et al. proposed using high-resolution UAV imagery as data to count the plant number of spinach. They uses machine vision—the Excess Green Index and Otsu's method—and transfer learning using convolutional neural networks to identify and count plants [5]. Azam Karami et al. proposed using a modified CenterNet architecture to identify and count maize plants in RGB images acquired from UAV. In their method, they also adopted transfer learning and few-shot learning to lift their performance by saving data [24]. Bruno T. Kitano et al. proposed using images captured via UAV assembled with an RGB sensor to count the plant number of corn. The method is a deep learning method and the network adopted is U-Net [24]. Javier Ribera et al. also used a CNN and RGB images obtained from UAV to count corn plants [25]. Mélissande Machefer et al. used a refitting strategy to fine tune the Mask-R-CNN using orthomosaic images captured via UAV. Their task focuses on two low-density crops, potato and lettuce. They demonstrated that using transfer learning on the Mask-R-CNN can significantly save data used for training [26]. Sungchan Oh et al. used the YOLOv3 and

unmanned aircraft systems data to count cotton plants [27]. Xiaodong Bai et al. proposed a new rice plant counting, locating, and sizing method named as RiceNet for rice plant counting. The data used in their work are also captured via UAV [28]. Hao Lu at al. proposed a network named as TasselNetV2+, also for rice counting. The data used in their work is high-Resolution RGB Imagery [4]. Hamza Mukhtar et al. proposed a method based on cross-consistency for the semantic segmentation of field images and an inception-based regression semi-supervised network for wheat plant counting. The data used in their experiments is obtained via UVA, which are high-resolution RGB imagery [29]. Lucas Prado Osco et al. proposed using a convolutional neural network approach for counting and geolocating citrus trees in UAV multispectral imagery. In their experiments, the results show that a better performance was obtained with the combination of green, red, and near-infrared bands [30].

There are some analogous work, such as counting the leaves and fruits. Sambuddha Ghosal et al. proposed a weakly supervised CNN for sorghum head detection and counting [31]. Nurulain Abd Mubin et al. used a deep CNN for oil palm tree detection and counting [32]. Hao Lu et al. proposed a network named as TasselNet and conducted counting maize tassels in the wild [33]. Shubhra Aich et al. investigated the problem of counting rosette leaves from an RGB image [34]. Jordan Ubbens et al. used deep CNN to count the leaves in rosette plants [35]. Md Mehedi Hasan et al. used CNN for the detection and analysis of wheat spikes [36]. In earlier times, Xiuliang Jin et al. estimated plant density of wheat crops by using SVM and very low altitude UAV imagery [37]. Maryam Rahnemoonfar et al. used Inception-ResNet to count the fruits and flowers [38].

By going through the existing works, we found that most of the works concerning plant counting now use deep learning methods; specifically, the semantic segmentation and OD methods. The main crops involved are predominantly cereal crops, such as corn, maize, wheat, etc. Another common characteristic is using RGB visible light imagery as materials.

### 2.2. Object Detection

The underlying problem of plant counting is the detection of the plants. In deep learning field, both OD and semantic segmentation can accomplish this task. The OD methods are more commonly used. OD is a computer vision task for identifying and localizing objects within an image or a video. The result of OD provides two pieces of information: the predicted locations and the predicted categories [39].

The OD has been significantly lifted by deep learning in recent years. There are anchor-free approaches and anchor-based approaches. The mainstream methods of the anchor-based approaches can be broadly categorized into two main types: one-stage approaches and two-stage approaches. The one-stage methods just conduct OD directly without any regions of interest (RoIs) proposals explicitly ahead. The image is divided into a grid and predicts the presence of objects and their bounding box coordinates at each grid cell. The representative works include YOLO (You Only Look Once) series, SSD (Single Shot MultiBox Detector), RetinaNet [40], etc. For the two-stage methods, they first propose potential RoIs of the image, then refine the features of these regions and finally classify them. Typical two-stage methods include R-CNN, Fast R-CNN, Faster R-CNN, Mask R-CNN, FPN, Libra R-CNN, etc. These methods typically achieve higher accuracy but are slower compared to one-stage methods [41,42].

YOLO is a series of OD networks. Different from the traditional OD methods that involve multiple stages, the networks of YOLO series unified the detection into a single neural network. After a single pass, the predicting bounding box coordinates and class probabilities are given directly. The common framework of YOLO introduces a ground-breaking approach to OD by formulating it as a regression problem. Getting benefits of the ont-stage passing, YOLO achieved real-time detection and obtained a balance between speed and accuracy [43]. For our specific purpose of plant counting, only one category is required to be detected and the real-time requirement is high. For the sake, YOLO is an

appropriate choice [44]. In this work, we choose the YOLOv8 that is the last version of the series.

Transfer learning is a technology in deep learning. It utilizes large-scale data in pre-training for obtaining the pre-trained model of a deep neural network, which is a start point of training in the target domain. Hence, it can be seen as a kind of training strategy. Instead of training from scratch, the pre-trained model already has basic or general knowledge which it gained from pre-training. By using transfer learning, the data demand is limited. For some applications that hardly collect enough data, transfer learning is beneficial for avoiding overfitting. Also, the generalization of the model is improved by the pre-training. Especially for a deeper learning neural network with more parameters, transfer learning helps save training data for the target domain [45].

## 3. Methods and Materials

The YOLO algorithms' series has become a widely used algorithm as a one-step algorithm for object detection [46]. In this study, we used YOLOV8 as the baseline network. The framework of YOLOv8 is as shown in Figure 1. YOLOv8 is a deep learning-based OD algorithm that enables fast, accurate, and robust OD and instance segmentation on high-resolution images. It includes three components: backbone part, neck part, and head part.

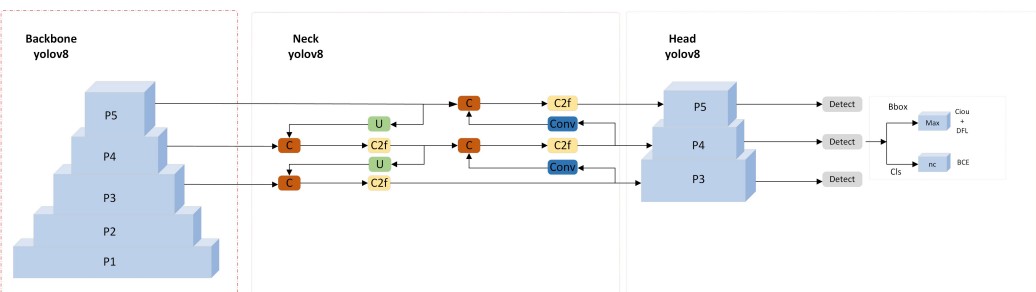

**Figure 1.** The framework of YOLOv8.

### 3.1. Backbone Network

The backbone network is a convolutional neural network used for extracting image features. YOLOv8 uses a similar backbone as YOLOv5 with some changes on the CSPLayer, now called the C2f module [43], which enhances gradient flow and feature fusion, thereby improving the feature representation capabilities. The C2f structure consists of multiple C2f modules, with each C2f module comprising two convolutional layers and a residual connection. The first convolutional layer reduces the number of channels, while the second convolutional layer restores the channel count.

C2f uses a $3 \times 3$ convolutional kernel as the first convolutional layer, allowing it to adapt the channel count differently for models of different scales, without following a fixed set of scaling factors. This enables the selection of an appropriate channel count based on the model's size, avoiding overfitting or underfitting. C2f employs a four-stage backbone network, with each stage consisting of 3-6-6-3 C2f modules, yielding feature maps of sizes $80 \times 80$, $40 \times 40$, $20 \times 20$, and $10 \times 10$, respectively. This allows for the extraction of multi-scale feature information to accommodate objects of various sizes.

In our work, we modify the first layer of the backbone network to adapt different input data. In addition to the traditional RGB images that has three input channels, our data has more diverse channel combinations. The visible RGB color images have three channels, as well as the red, green, and near-infrared combination images also have three channels. The single channel inputs include the narrow red band (NR), narrow green band (NG), near-infrared band (NIR), and red edge band (RE). In addition, all of the bands are combined together into a seven-band combination. Therefore, the structure of the first layer has to be modified to fit the input data, as shown in Figure 2.

In order to avoid overfitting, we adopt transfer learning and load the pre-trained weights before training with our target domain data. Since the structure of the YOLOv8 is modified, the weights cannot entirely match the modified structure. We only load the rest weights of the trained model except for the first convolutional layer.

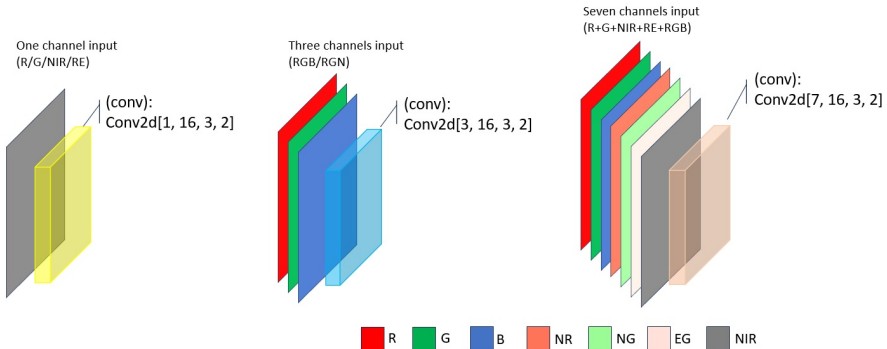

**Figure 2.** The modified first layer of YOLOv8 for adapting the different inputs.

*3.2. Neck Network*

The neck network is a network used to fuse feature maps of different scales. YOLOv8 uses a PANet structure, which can achieve top-down and bottom-up feature fusion, increasing receptive fields and semantic information. The spacial pyramid pooling layer in the SPPF module can use multiple convolution kernels of different sizes to perform pooling operations to extract multi-scale features.

YOLOv8 uses two mechanisms in the neck network: one is a feature fusion mechanism and the other is a feature selection mechanism. YOLOv8 uses a new feature fusion mechanism, which can dynamically adjust the fusion weight according to feature maps of different scales to improve the fusion effect. This fusion mechanism uses an attention module that can calculate the similarity between feature maps at different scales and assign fusion weights based on the similarity. YOLOv8 also uses a new feature selection mechanism, which can select appropriate feature maps as the output according to the needs of different tasks. This selection mechanism uses a classifier that predicts the contribution of each feature map based on the task label and selects the optimal feature map based on the contribution.

*3.3. Head Network*

The head network is a network used to predict the target category and location. YOLOv8 uses a new decoupled head structure, which can separate classification and regression tasks, improving model flexibility and stability. The decoupled head structure consists of two branches: one is the classification branch, which is used to predict the category probability of each pixel, and the other is the regression branch, which is used to predict the bounding box coordinates of each pixel.

YOLOv8 also uses a new integral form representation, which can model the bounding box coordinates into a probability distribution, improving regression accuracy and robustness. The integral form representation consists of two convolutional layers and a softmax layer, which can output the offset probability of each pixel in four directions and obtain the final bounding box coordinates via an integral operation. After obtaining the bounding box coordinates, YOLOv8 uses the loss function of CIoU [47] and DFL [48] for boundary box loss and binary cross entropy for classification loss. It is a new positive and negative sample matching strategy, which can determine positive and negative samples according to the IoU (intersection over union) between the boundary box and the real label. Specifically, if the IoU between the bounding box generated by a pixel and the real label is greater than or equal to 0.5, the pixel and bounding box are regarded as positive samples; if it is less than 0.4, they are regarded as negative samples; and if it is within 0.4 and 0.5, they are ignored.

### *3.4. Hardware*

The configuration hardware in this work is as follows: a server was equipped with an Intel Xeon Platinum 8336C processor running at 2.30 GHz, featuring 64 logical cores, and it was also outfitted with 128 GB of Samsung DDR4-2400 ECC RDIMM memory, consisting of 16 modules, each with an 8 GB capacity.

The UVA used in this work is equipped with an RGB camera and a multispectral camera. The RGB camera captures the RGB images and the multispectral camera with a narrow band filter captures four multispectral images. The multispectral images include NG (560 ± 16 nm), NR (650 ± 16 nm), RE (730 ± 16 nm), and NIR (860 ± 26 nm).

### *3.5. Data and Data Pre-Processing*

#### 3.5.1. Data

In the field cultivation season in 2023, we captured multispectral images five times at the location (25°01′32.0″ N 103°59′21.4″ E). The flight altitude was set at 15 m. Starting from May, when the seedlings were transplanted in the field, to August, when the leaves were ready for harvesting, we visited the field approximately every 15 days to collect images of the tobacco plants at various growth stages. A comprehensive dataset is essential to ensure the model's robustness. Additionally, continuous monitoring of the tobacco plant's growth process is valuable for harvest estimation. Plants in different growth stages exhibit distinct appearances and sizes in the orthophoto imagery

As shown in Figure 3, Figure 3a displays the RGB image captured on 2 July. It is evident that the tobacco plants were quite small immediately after transplantation into the field, with significant gaps between them, and the black plastic film was still present around the plants. As the growth process continued, the plants grew larger and the gaps between them gradually disappeared. By the arrival of August, the tobacco plants entered the maturity phase, as indicated by the yellowing of lower leaves, signaling their readiness for harvest. Figure 3e shows the plants after pruning and the first harvest of ripe leaves.

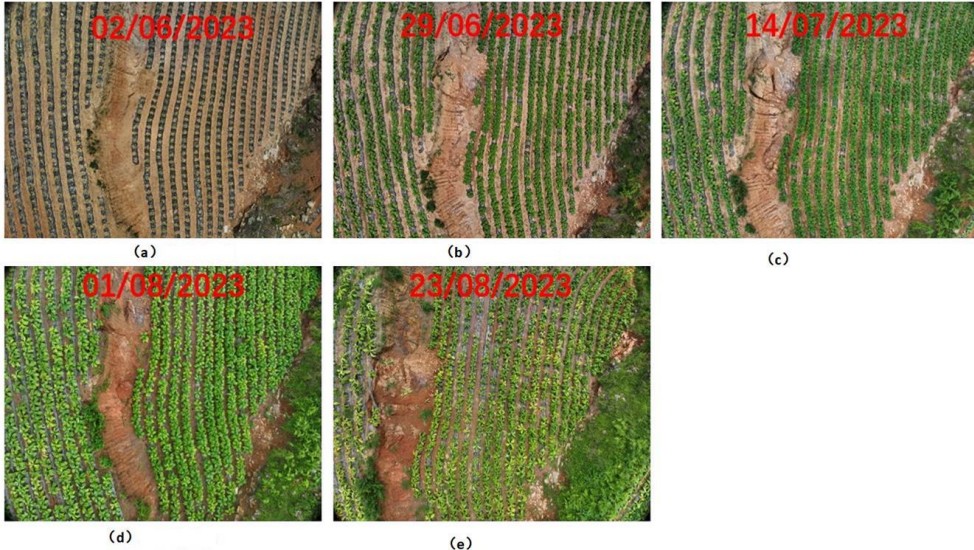

**Figure 3.** The five different growth stages of tobacco plants in the field.

We choose a site for our experiments. The view of the site is as shown in Figure 4. In the site, except for the tobacco plant field, there are many other types of land surfaces, such as lanes, gourds, stones, trees, weeds, and other plants. In the field, corn and soy bean are also planted alternatively. The data of a complex land cover is good to see the robustness of our method.

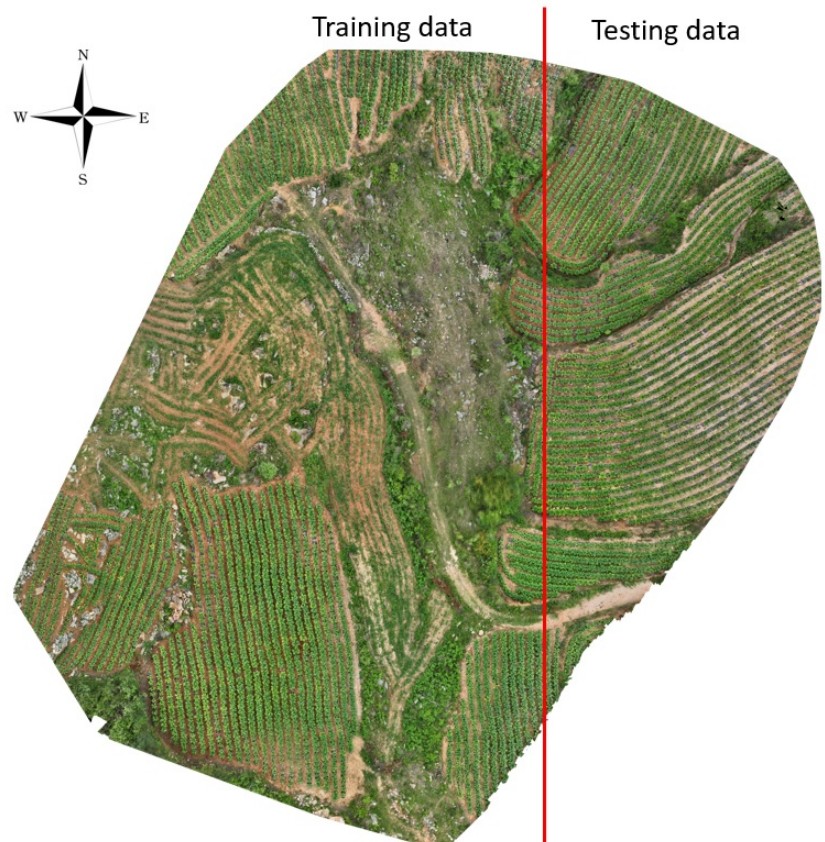

**Figure 4.** The view of the plot we observed. The left part of the red line is used for training, and the right part is used for testing. The ratio of the two parts is 8:2.

Each photograph captured resulted in five different spectral images, which are the RGB image, NG image, NR image, RE image, and NIR image, as shown in Figure 5. For this site, 2315 shots are captured by the UAV, with 463 shots for each. These shots of each kind of spectrum are stitched together into a complete image of the site. After stitching, we keep the intersection area of the five spectrum images for making sure that the each channel does not include an empty area inside.

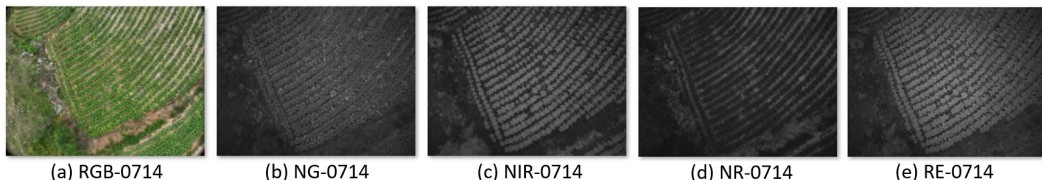

(a) RGB-0714    (b) NG-0714    (c) NIR-0714    (d) NR-0714    (e) RE-0714

**Figure 5.** The five different spectrum images of 14 July 2023. (**a**) is the RGB visible light image; (**b**) is the narrow green band image; (**c**) is the Near-Infrared band image; (**d**) is the narrow red band image; and (**e**) is the red edge band image.

3.5.2. Data Pre-Processing

For the site, we observed 503 shots. In order to utilize more bands except for the RGB, we merged the RGB, NR, NG, RE, and NIR together, as shown in Figure 6a. Another combination of bands is G, R, and NIR, as shown in Figure 6c. It is a commonly used combination in agriculture to detect vegetation. Due to the two cameras being located at different places of the UAV, the RGB image and the other multispectral images of the same shot still have slight bias. Therefore, the coordinates are firstly calibrated in merging stage.

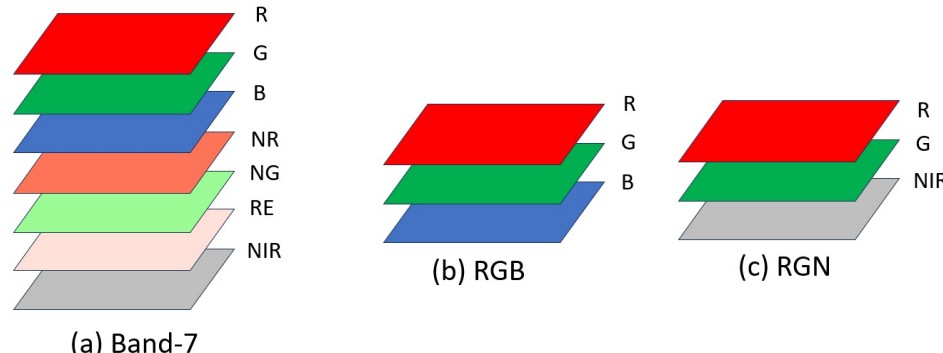

**Figure 6.** The illustrations of different spectrum band combinations. (**a**) is a combination of the full seven bands; (**b**) is the three-band combination of visible light; and (**c**) is the three-band combination of R, G, and NIR.

The size of the image after stitching is too large to feed into the deep learning neural network. Hence, before training, the stitched image is segmented into small patches (640 × 640). The plants in the images are labeled by using Labelme.

## 4. Experiments and Results

In total, we conducted 103 experiments. We adopted the relevant settings of the basic network (YOLOv8), including the settings of basic parameters. Through a series of experiments and comparisons, we found that when the batch of the original basic network is set to 128 and epoch is set to 200, the model performs best and has smaller fluctuations. In view of this, we adjusted the batch and epoch to better fit our multispectral dataset while keeping the other parameters of the base network unchanged. These experiments can be approximately summarized into three categories: the performance, the accuracy of plant counting in slices, and the accuracy of plant counting in a big plot.

### 4.1. Metrics

In this work, we use three common metrics to evaluate the performance: precision, recall, and mAP [49]. A critical metric is the accuracy of plant counting. Before defining these metrics, the labels of true positive (*TP*), False Positive (*FP*), False Negative (*FN*), and True Negative (*TN*) must be defined. In OD, TP refers to a correct detection when the predicted class is matched and the IoU is greater than a threshold. FP refers to a wrong detection when the class is unmatched or the detection with IoU is less than the threshold. FN is the number of missed detections of positive instances. TN does not apply in OD. In this work, because there is only one detection class, tobacco plant, only positive has meaning and negative has non-meaning. TP indicates that the tobacco plant is detected as tobacco plant. FP indicates that the other things are detected as tobacco plant.

The precision is calculated by dividing the number of TP by the total number of positive detections, which is represented in Formula (1). It indicates the ability of a model to identify only the relevant objects and also has insights into the model's ability to avoid false positive detections. A high precision indicates a low rate of incorrectly identifying the background or non-target objects as positive detections.

$$Precision = \frac{TP}{TP + FP} = \frac{TP}{All\ detections} \tag{1}$$

The second metric is recall. It is the ability of a model to find all the relevant cases. It is calculated by dividing the number of true positive detections by the sum of true positive detections and false negative detections, which is represented in Formula (2). It indicates the proportion of correctly identified positive instances out of all the actual

positive instances of a given dataset. A high recall value indicates that the model has a low rate of missing relevant objects and is effective at capturing most of the positive instances.

$$Recall = \frac{TP}{TP + FN} = \frac{TP}{P} = \frac{TP}{All\ ground\ truths} \tag{2}$$

In Formula (2), the recall value is the ratio of the correct detection number of tobacco plants to the number of all ground truths. However, it does not indicate the plant counting accuracy. In here, it is calculated as an average value with ten different IoU settings from 0.50 to 0.95 and with a space of 0.05. It still indicates the counting accuracy to a certain degree.

mAP50 is another staple metric used for project detection. It is a variation in the mean average precision (mAP) metric specifically calculated, at a fixed threshold of intersection over a union, as 0.5. The threshold of 0.5 is commonly used to determine whether a detection is considered a true positive or a false positive. For every category, the average precision (AP) is calculated as the area under the precision–recall curve. Then, the mAP is the mean value of the AP of all categories. For our specific task, the number of categories is one because we only have one category that is tobacco plant.

Except for the three metrics commonly used for OD, the plant counting accuracy is the most important metric we considered, which is described in Formula (3). It is the ratio of the number of predicted stands divided by the actual number of stands.

$$Plant\ counting\ accuracy = \frac{Predicted\ stands}{Actual\ stands} \tag{3}$$

### 4.2. Experiment of Different Bands

The group of experiments aims to explore the performance of different spectrum combinations. Except for the five kind of images obtained from UAV, Band-7 and RGN introduced in Section 3.5.2 are tested in our experiments.

Each experiment listed in Table 1 was conducted within the images captured in one flight. The tobacco seedling images used in the experiment were obtained after planting on 29 April 2023. Images from e1 to e7 were captured 30 days after the tobacco seedlings were planted (on 2 June 2023). Images from e8 to e14 were taken 57 days after planting (on 29 June 2023). Images from e15 to e21 were captured 72 days after planting (on 14 July 2023). Images from e22 to e28 were taken 90 days after planting (on 1 August 2023). Images from e29 to e35 were captured 112 days after planting (on 23 August 2023). With the segmented images from the complete image, 80% of the images are used for training and the remaining 20% are used for testing, as shown in Figure 4. The column data indicates the date that the images were taken and the experiments are the five groups according to the date.

Firstly, we can see that the RGN setting gains the best performance in every group. The images with more channels perform better then these single-channel images. The reason is that more channels always contain more information. Comparing the four single channels, R, G, RE, and NIR, we find that the RGN combination performs the best. Although the Band-7 includes seven channels that cover all of the above bands, it does not show the best results. The reason is the band-7 contain redundant information and causes the network to converge slowly or not converge even at the lowest point. The red, green, and NIR are three spectral bands which are more sensitive. Comparing the different data, we find that 0801 is the lowest group. The reason is that in this stage, the plants are the biggest in their life cycle, where the gap between plants are closed and the plants start joining together. It causes the difficulty of tobacco plant instance detection. The detection results are better in group 0823 after that because the first-round mature leaves were reaped, resulting in the plants not getting so close to each other anymore.

**Table 1.** The results of different band combinations on different dates.

| ID | Training Data | Testing Data | Channels | Precision (%) | Recall (%) | mAP50 (%) | mAP50-95 (%) |
|---|---|---|---|---|---|---|---|
| e1 | NR-0602-316 | NR-0602-80 | 1 | 94.43 | 96.17 | 98.02 | 53.40 |
| e2 | NG-0602-316 | NG-0602-80 | 1 | 93.66 | 92.09 | 95.44 | 49.78 |
| e3 | RE-0602-316 | RE-0602-80 | 1 | 90.69 | 92.15 | 93.40 | 50.22 |
| e4 | NIR-0602-316 | NIR-0602-80 | 1 | 95.80 | 95.04 | 97.16 | 58.98 |
| e5 | RGB-0602-316 | RGB-0602-80 | 3 | 96.56 | 98.48 | 99.04 | 70.00 |
| e6 | RGN-0602-316 | RGN-0602-80 | 3 | 97.10 | 98.87 | 99.14 | 70.79 |
| e7 | Band-7-0602-316 | Band-7-0602-80 | 7 | 96.37 | 95.32 | 98.27 | 66.13 |
| e8 | NR-0629-460 | NR-0629-115 | 1 | 97.32 | 97.18 | 99.26 | 75.23 |
| e9 | NG-0629-460 | NG-0629-115 | 1 | 96.95 | 98.12 | 99.32 | 74.66 |
| e10 | RE-0629-460 | RE-0629-115 | 1 | 97.65 | 98.35 | 99.20 | 76.96 |
| e11 | NIR-0629-460 | NIR-0629-115 | 1 | 97.02 | 98.40 | 99.29 | 76.85 |
| e12 | RGB-0629-460 | RGB-0629-115 | 3 | 97.90 | 98.57 | 99.37 | 81.38 |
| e13 | RGN-0629-460 | RGN-0629-115 | 3 | 98.30 | 98.83 | 99.42 | 83.77 |
| e14 | Band-7-0629-460 | Band-7-0629-115 | 7 | 97.38 | 98.36 | 99.38 | 82.03 |
| e15 | NR-0714-304 | NR-0714-76 | 1 | 96.07 | 90.61 | 96.20 | 63.45 |
| e16 | NG-0714-304 | NG-0714-76 | 1 | 97.59 | 84.50 | 91.79 | 64.30 |
| e17 | RE-0714-304 | RE-0714-76 | 1 | 97.58 | 88.51 | 94.24 | 72.06 |
| e18 | NIR-0714-304 | NIR-0714-76 | 1 | 94.70 | 88.32 | 94.38 | 67.60 |
| e19 | RGB-0714-304 | RGB-0714-76 | 3 | 96.35 | 87.53 | 89.14 | 71.14 |
| e20 | RGN-0714-304 | RGN-0714-76 | 3 | 98.80 | 94.51 | 98.36 | 74.05 |
| e21 | Band-7-0714-304 | Band-7-0714-76 | 7 | 98.03 | 89.73 | 93.83 | 70.21 |
| e22 | NR-0801-352 | NR-0801-88 | 1 | 85.29 | 69.35 | 71.06 | 42.40 |
| e23 | NG-0801-352 | NG-0801-88 | 1 | 86.80 | 78.06 | 77.54 | 47.13 |
| e24 | RE-0801-352 | RE-0801-88 | 1 | 86.06 | 78.39 | 75.40 | 46.18 |
| e25 | NIR-0801-352 | NIR-0801-88 | 1 | 85.21 | 76.88 | 75.38 | 46.59 |
| e26 | RGB-0801-352 | RGB-0801-88 | 3 | 82.09 | 77.84 | 72.56 | 45.67 |
| e27 | RGN-0801-352 | RGN-0801-88 | 3 | 87.56 | 78.73 | 75.73 | 47.62 |
| e28 | Band-7-0801-352 | Band-7-0801-88 | 7 | 85.72 | 75.18 | 73.80 | 45.65 |
| e29 | NR-0823-386 | NR-0823-97 | 1 | 95.69 | 89.32 | 95.63 | 65.51 |
| e30 | NG-0823-386 | NG-0823-97 | 1 | 95.40 | 95.57 | 97.80 | 72.80 |
| e31 | RE-0823-386 | RE-0823-97 | 1 | 93.62 | 95.13 | 96.97 | 71.97 |
| e32 | NIR-0823-386 | NIR-0823-97 | 1 | 93.26 | 95.56 | 97.54 | 71.35 |
| e33 | RGB-0823-386 | RGB-0823-97 | 3 | 95.12 | 91.12 | 93.04 | 71.16 |
| e34 | RGN-0823-386 | RGN-0823-97 | 3 | 97.42 | 95.76 | 98.53 | 75.67 |
| e35 | Band-7-0823-386 | Band-7-0823-97 | 7 | 94.58 | 95.84 | 98.00 | 70.70 |

The format of training data and testing data is bands-date-number of images of data. Channels indicate the number of channels of input data. The red color font indicates the best result and the blue color font indicates the second best result.

The plant counting is another critical result of our task. As mentioned before, a right detection is decided by both the confidence and IoU. In the group of counting experiments from e36 to e70, the threshold of confidence is set as 0.25 and the threshold of IoU is set as 0.5. The data used in this group follow the same setting in Table 1. The GT of plants is the number of plants of the total testing images. The plants detected is the sum of plant numbers detected in the testing set. The counting results are reported in Table 2. The results show that the RGN band combinations still keep the best results of plant counting. By using the slice images, all of the counting accuracy using different band combinations are kept in the level higher than 95%.

**Table 2.** The plant counting results of different band combinations on different dates.

| ID | Training Data | Testing Data | Channels | GT of Plants | Plants Detected | ACC (%) |
|----|---------------|--------------|----------|--------------|-----------------|---------|
| e36 | NR-0602-316 | NR-0602-80 | 1 | 575 | 559 | 97.21 |
| e37 | NG-0602-316 | NG-0602-80 | 1 | 575 | 552 | 96.60 |
| e38 | RE-0602-316 | RE-0602-80 | 1 | 575 | 558 | 97.04 |
| e39 | NIR-0602-316 | NIR-0602-80 | 1 | 575 | 551 | 95.82 |
| e40 | RGB-0602-316 | RGB-0602-80 | 3 | 575 | 562 | 97.73 |
| e41 | RGN-0602-316 | RGN-0602-80 | 3 | 575 | 569 | 98.95 |
| e42 | Band-7-0602-316 | Band-7-0602-80 | 7 | 575 | 553 | 96.17 |
| e43 | NR-0629-460 | NR-0629-115 | 1 | 556 | 543 | 97.66 |
| e44 | NG-0629-460 | NG-0629-115 | 1 | 556 | 541 | 97.30 |
| e45 | RE-0629-460 | RE-0629-115 | 1 | 556 | 540 | 97.12 |
| e46 | NIR-0629-460 | NIR-0629-115 | 1 | 556 | 533 | 95.86 |
| e47 | RGB-0629-460 | RGB-0629-115 | 3 | 556 | 545 | 98.02 |
| e48 | RGN-0629-460 | RGN-0629-115 | 3 | 556 | 549 | 98.74 |
| e49 | Band-7-0629-460 | Band-7-0629-115 | 7 | 556 | 536 | 96.40 |
| e50 | NR-0714-304 | NR-0714-76 | 1 | 770 | 753 | 97.79 |
| e51 | NG-0714-304 | NG-0714-76 | 1 | 770 | 747 | 97.01 |
| e52 | RE-0714-304 | RE-0714-76 | 1 | 770 | 756 | 98.18 |
| e53 | NIR-0714-304 | NIR-0714-76 | 1 | 770 | 739 | 95.97 |
| e54 | RGB-0714-304 | RGB-0714-76 | 3 | 770 | 758 | 98.44 |
| e55 | RGN-0714-304 | RGN-0714-76 | 3 | 770 | 762 | 98.96 |
| e56 | Band-7-0714-304 | Band-7-0714-76 | 7 | 770 | 755 | 98.80 |
| e57 | NR-0801-352 | NR-0801-88 | 1 | 652 | 617 | 94.63 |
| e58 | NG-0801-352 | NG-0801-88 | 1 | 652 | 629 | 96.47 |
| e59 | RE-0801-352 | RE-0801-88 | 1 | 652 | 618 | 94.78 |
| e60 | NIR-0801-352 | NIR-0801-88 | 1 | 652 | 611 | 93.71 |
| e61 | RGB-0801-352 | RGB-0801-88 | 3 | 652 | 620 | 96.09 |
| e62 | RGN-0801-352 | RGN-0801-88 | 3 | 652 | 636 | 97.54 |
| e63 | Band-7-0801-352 | Band-7-0801-88 | 7 | 652 | 615 | 94.32 |
| e64 | NR-0823-386 | NR-0823-97 | 1 | 612 | 593 | 96.89 |
| e65 | NG-0823-386 | NG-0823-97 | 1 | 612 | 589 | 96.24 |
| e66 | RE-0823-386 | RE-0823-97 | 1 | 612 | 591 | 96.56 |
| e67 | NIR-0823-386 | NIR-0823-97 | 1 | 612 | 585 | 95.58 |
| e68 | RGB-0823-386 | RGB-0823-97 | 3 | 612 | 591 | 96.56 |
| e69 | RGN-0823-386 | RGN-0823-97 | 3 | 612 | 601 | 98.20 |
| e70 | Band-7-0823-386 | Band-7-0823-97 | 7 | 612 | 587 | 95.91 |

The format of training data and testing data is bands-date-number of images of data. Channels indicate the number of channels of input data. The red color font indicates the best result and the blue color font indicates the second best result.

### 4.3. The Setting of Confidence

The confidence score is a possibility associated with each detected object. It represents how accurately and confidently it belongs to the predicted category. As shown in Figure 7, the value on top of the detection bounding box is the confidence value of the object. Since we only have one category in the detection task, the confidence indicates that the possibility of the detection is tobacco plant. We can see that generally, the detection possibilities of a tobacco plant located in the middle of the image are over 50% because they keep their entire shape. Even plants at the border of the image can still be identified, just with a lower confidence. It indicates that the YOLOv8 is powerful for OD. We conducted a group of experiments to show the relationship of confidence with the precision and recall, as shown in Table 3 and Figure 8. e71, e72, and e73 all have a good balance of the precision of recall. For the plant counting task, because the recall value is more critical, in the rest experiments, the confidence is set as 0.25. Generally, when the confidence is smaller than 0.2, the tobacco plant is only a very small part at the border of the image and will not be counted.

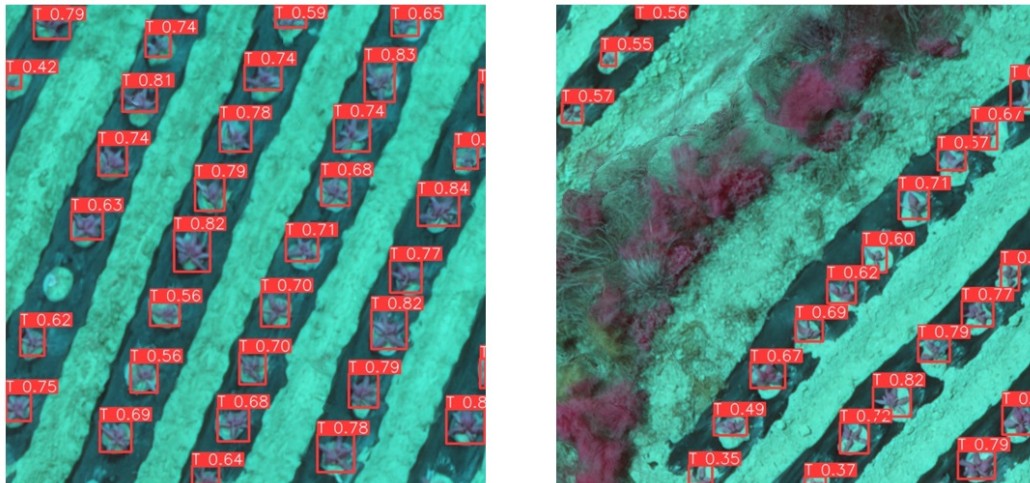

**Figure 7.** Examples of detection with the detection confidence. The images are RGN.

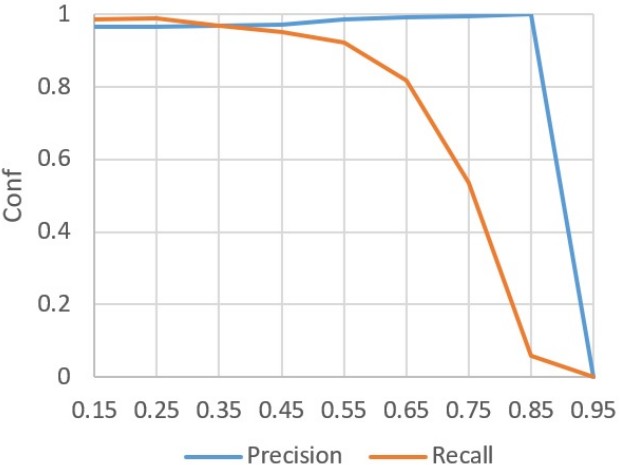

**Figure 8.** Graph of the relationship between confidence, precision and recall.

**Table 3.** The relationship of the confidence value with the precision and recall.

| ID | Confidence | Precision (%) | Recall (%) |
|----|-----------|---------------|-----------|
| e71 | 0.15 | 96.71 | 98.73 |
| e72 | 0.25 | 96.50 | 98.94 |
| e73 | 0.35 | 97.00 | 97.10 |
| e74 | 0.45 | 97.29 | 95.32 |
| e75 | 0.55 | 98.80 | 92.18 |
| e76 | 0.65 | 99.33 | 81.68 |
| e77 | 0.75 | 99.59 | 53.54 |
| e78 | 0.85 | 100.00 | 5.92 |
| e79 | 0.95 | 0 | 0 |

The red color font indicates the best result and the blue color font indicates the second best result.

### 4.4. Experiment of Mixed Data of Five Dates

In the previous experiments, we conduct each training and testing on the same date. A model with good robustness expectations is better for adapting into different periods. For this sake, we conducted a group of experiments with mixed data, from e80 to e86. We use 80% of all the data of five dates for training, and 20% of all the data of five dates for testing. The results are shown in Table 4. We can see that these metrics still keep at a high level. The plant counting is also conducted with the mixed data setting. As shown in Table 5,

e87 to e93 follow the same data setting of e80 to e86, respectively. The results show that training the network with the mixed data allows the model to adapt to different periods of detection.

**Table 4.** The results of the different bands' comparison, predicted trees, and real plans.

| ID | Training Data | Testing Data | Channels | Precision (%) | Recall (%) | mAP50 (%) | mAP50-95 (%) |
|----|---------------|--------------|----------|---------------|------------|-----------|--------------|
| e80 | NR-five dates-1818 | NR-five dates-456 | 1 | 94.06 | 90.00 | 92.34 | 63.31 |
| e81 | NG-five dates-1818 | NG-five dates-456 | 1 | 93.77 | 86.13 | 90.13 | 63.45 |
| e82 | RE-five dates-1818 | RE-five dates-456 | 1 | 93.47 | 88.24 | 92.13 | 64.94 |
| e83 | NIR-five dates-1818 | NIR-five dates-456 | 1 | 94.21 | 87.46 | 90.13 | 64.75 |
| e84 | RGB-five dates-1818 | RGB-five dates-456 | 3 | 94.23 | 92.14 | 88.90 | 68.25 |
| e85 | RGN-five dates-1818 | RGN-five dates-456 | 3 | 95.66 | 90.30 | 93.67 | 70.98 |
| e86 | Band-7-five dates-1818 | Band-7-five dates-456 | 7 | 94.06 | 76.79 | 86.23 | 65.21 |

The format of training data and testing data is bands-date-number of images of data. Channels indicate the number of channels of input data. The red color font indicates the best result and the blue color font indicates the second best result.

**Table 5.** The plants counting results of different bands combination with five dates data together.

| ID | Training Data | Testing Data | Channels | GT of Plants | Plants Detected | ACC (%) |
|----|---------------|--------------|----------|--------------|-----------------|---------|
| e87 | NR-five dates-1818 | NR-five dates-456 | 1 | 3165 | 3109 | 98.23 |
| e88 | NG-five dates-1818 | NG-five dates-456 | 1 | 3165 | 3111 | 98.29 |
| e89 | RE-five dates-1818 | RE-five dates-456 | 1 | 3165 | 3108 | 98.20 |
| e90 | NIR-five dates-1818 | NIR-five dates-456 | 1 | 3165 | 3101 | 97.98 |
| e91 | RGB-five dates-1818 | RGB-five dates-456 | 3 | 3165 | 3121 | 98.60 |
| e92 | RGN-five dates-1818 | RGN-five dates-456 | 3 | 3165 | 3129 | 98.86 |
| e93 | Band-7-five dates-1818 | Band-7-five dates-456 | 7 | 3165 | 3120 | 98.57 |

The format of training data and testing data is bands-date-number of images of data. Channels indicate the number of channels of input data. The red color font indicates the best result and the blue color font indicates the second best result.

### 4.5. Experiment Test of the Whole Land

In the previous experiments, we conducted the testing experiments with the images that were made into pieces. Also, the counting results are reported as a sum value of all pieces of the images. From the perspective of an application, users are more interested in knowing the number of plants in a given field. For this purpose, we design another experiment testing with a whole piece of field. We choose a plot named as plot-s9 from the testing part, as shown in Figure 9a. It can be sliced into nine slices of a 640 × 640 size. The reason behind using nine slices is they covered the cases of the left, right, top, and bottom neighboring.

In training, the left part is still split into pieces of a small size (640 × 640). In testing, it is not reasonable to feed the entire image into the neural network. No matter if the image is resized or kept at the initial size, the detection accuracy is degraded. When the slice has the same size as the training image, the detection accuracy is the highest. However, the parts of the plant located at the edges of the images are detected as instances. One plant is cut into two parts that are detected in the neighboring images. If a plant is segmented into two slices, they are probably detected as two instances. For a whole piece of land, it is not accurate anymore to count the number of plants just via a simple summation because the plants at the border of the slices may be counted twice. Hence, we design an algorithm to count the plants of a big plot of land. As shown in Figure 10a, a tobacco plant sitting at the cutting line of two slices is separated into two parts, which are either half and half or a big part with a small part. The YOLOv8 is powerful enough to detect the plant even if is just a small part. When putting the slices together, the plant will be counted twice because it is detected twice in the two slices and the two detected bounding boxes with non-intersection, respectively. It is hard to tell if one plant was detected twice or if two plants were detected separately.

Aiming at the problem, in testing, the entire image for testing is split into slices, with an overlap area. There are two benefits of segmenting the image in this way. Firstly, if a plant coincidentally is located at the slicing line, the overlap slicing makes each of the parts larger and guarantees a higher detection confidence. Secondly, when putting the detection results of all slices together, the detected bounding boxes of the two parts from one plant are overlapped, as shown in Figure 10b. If it happens, only one plant is counted. We conducted two groups of experiments to evaluate this post-processing algorithm. In Figure 9a, it shows plot-s9 with 1800 × 1800 pixels. The slices covered by a red or green color are 640 × 640. The width of the overlap area is set as 60 pixels. Then, the images are split into nine slices, which are then fed into the network.

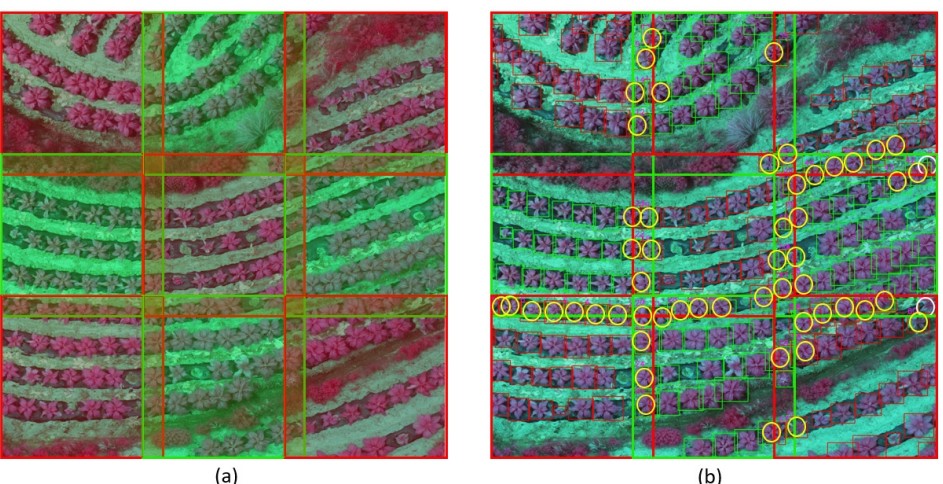

**Figure 9.** The illustration of the plant counting of plot-s9 in 29 June 2023. (**a**) is the slicing illustration of plot-s9. This plot is sliced into nine slices. (**b**) is the detection results. The red and green bounding boxes are the detection results in the independent slices. The yellow circle represents that the plant is detected twice in the neighboring slices.

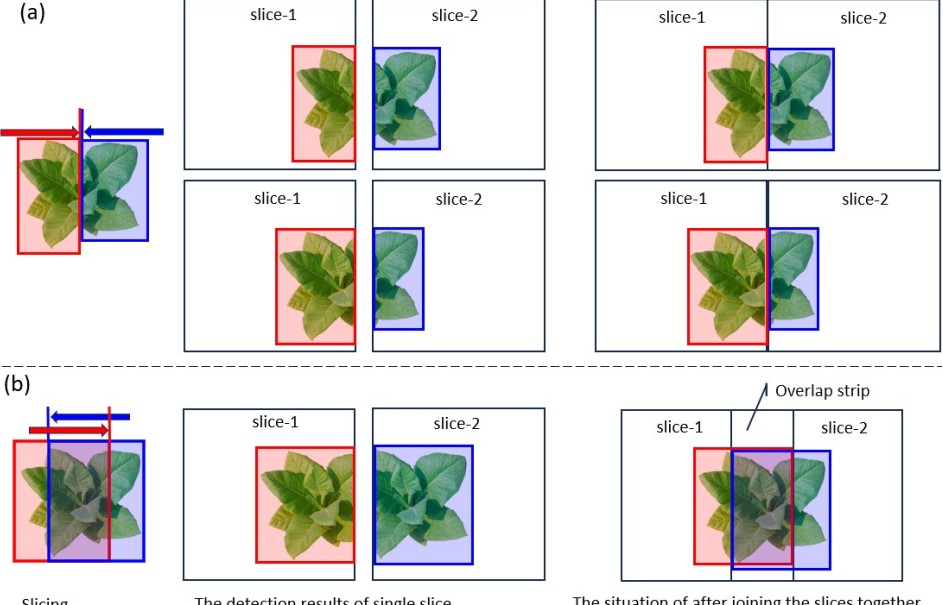

**Figure 10.** The illustration of slicing. (**a**) shows two cases of slicing without an overlap area. (**b**) shows the case of slicing with an overlap area. The red and blue boxes represent two different pictures. The red and blue arrows indicate that there is a common tobacco plant in the two pictures, but this tobacco plant is cut in half, and we need to overlap the same tobacco plant in the two different pictures in the direction of the arrow.

As shown in Table 6, e94 to e98 use the model trained by the data on the same date as the testing data. Taking e95 as an example, the sum of the detected plants is 272. The results are the same as the GT-1. However, the true plants in plot-s9 are 233. Obviously, the plants located in the overlap strips are repeatedly counted. The counting algorithm we designed only counts once if two detected bounding boxes are overlapped. As shown in Figure 9b, the plants marked with yellow circles are supposed to be one but detected twice in two slices at the same time. They are marked because the two or more detected bounding boxes are overlapped. There are 43 plants that are counted repeatedly. After noting the repeated counting, the final counting is 229. After the post-processing, the detection accuracy is high as 98.28%. We found that e94 gained the best counting accuracy and e95 has the second best counting accuracy. Then, the accuracy reduced along the time line. The reason is that the tobacco plants grew bigger along the time line and overlapped. Hence, the detection accuracy of the plant instance are effected. However, even in the last date of e98, the results still stay at a high-level as 95.65%.

**Table 6.** The plant counting results of plot-s9 in different dates.

| ID | Date | GT-1 | Detected-1 | Detected-o | GT-2 | Detected-2 | ACC (%) |
|----|------|------|------------|------------|------|------------|---------|
| e94 | 0602 | 272 | 272 | 43 | 233 | 229 | <span style="color:red">98.28</span> |
| e95 | 0629 | 265 | 271 | 65 | 210 | 206 | <span style="color:blue">98.10</span> |
| e96 | 0714 | 290 | 298 | 70 | 234 | 228 | 97.44 |
| e97 | 0801 | 283 | 289 | 76 | 220 | 213 | 96.82 |
| e98 | 0823 | 289 | 301 | 81 | 230 | 220 | 95.65 |

The data used in this group of the experiment is the RGN band combination. The training data and testing data in an experiment belongs to the same date. GT-1 is the sum of the GT of the nine slices. Detected-1 is the sum of the detected plants of the nine slices. Detected-o is the number of plants detected more than once. GT-2 is the true number of plants in plot-s9. Detected-2 is the results calculated by using our post-processing algorithm. The red color font indicates the best result and the blue color font indicates the second best result.

e99 to e103 is another group of the experiment using the same model trained with the data of all five dates, where the testing data is plot-s9. The results are as shown in Table 7 and Figure 11. In this group, we found that the plants' counting accuracy is even higher than the results shown in Table 6. The reason is that the model is trained by a more comprehensive dataset. In the field, even the plants are also always presented as either larger or smaller. Using a model that trained only the data of a certain period, the robustness of the model is reduced. As shown in Figure 11, the plants in the region of non-intersection are detected accurately, and the plants located in the overlap strips are also processed via the post-processing algorithm.

**Table 7.** The detection results of a big plot of land.

| ID | Date | GT-1 | Detected-1 | Detected-o | GT-2 | Detected-2 | ACC (%) |
|----|------|------|------------|------------|------|------------|---------|
| e99 | 0602 | 272 | 277 | 46 | 233 | 231 | <span style="color:red">99.14</span> |
| e100 | 0629 | 265 | 271 | 64 | 210 | 207 | <span style="color:blue">98.57</span> |
| e101 | 0714 | 290 | 307 | 77 | 234 | 230 | 98.29 |
| e102 | 0801 | 283 | 290 | 74 | 220 | 216 | 98.18 |
| e103 | 0823 | 289 | 306 | 83 | 230 | 223 | 96.96 |

The data used in this group of the experiment is the RGN band combination. In this experiment, the training data is formed by taking 80% of the data from five different time periods and combining them, while the testing data consists of 20% of the data from their respective time periods. GT-1 is the sum of the GT of the nine slices. Detected-1 is the sum of the detected plants of the nine slices. Detected-o is the number of plants detected more than once. GT-2 is the true number of plants in plot-s9. Detected-2 is the results calculated by using our post-processing algorithm. The red color font indicates the best result and the blue color font indicates the second best result.

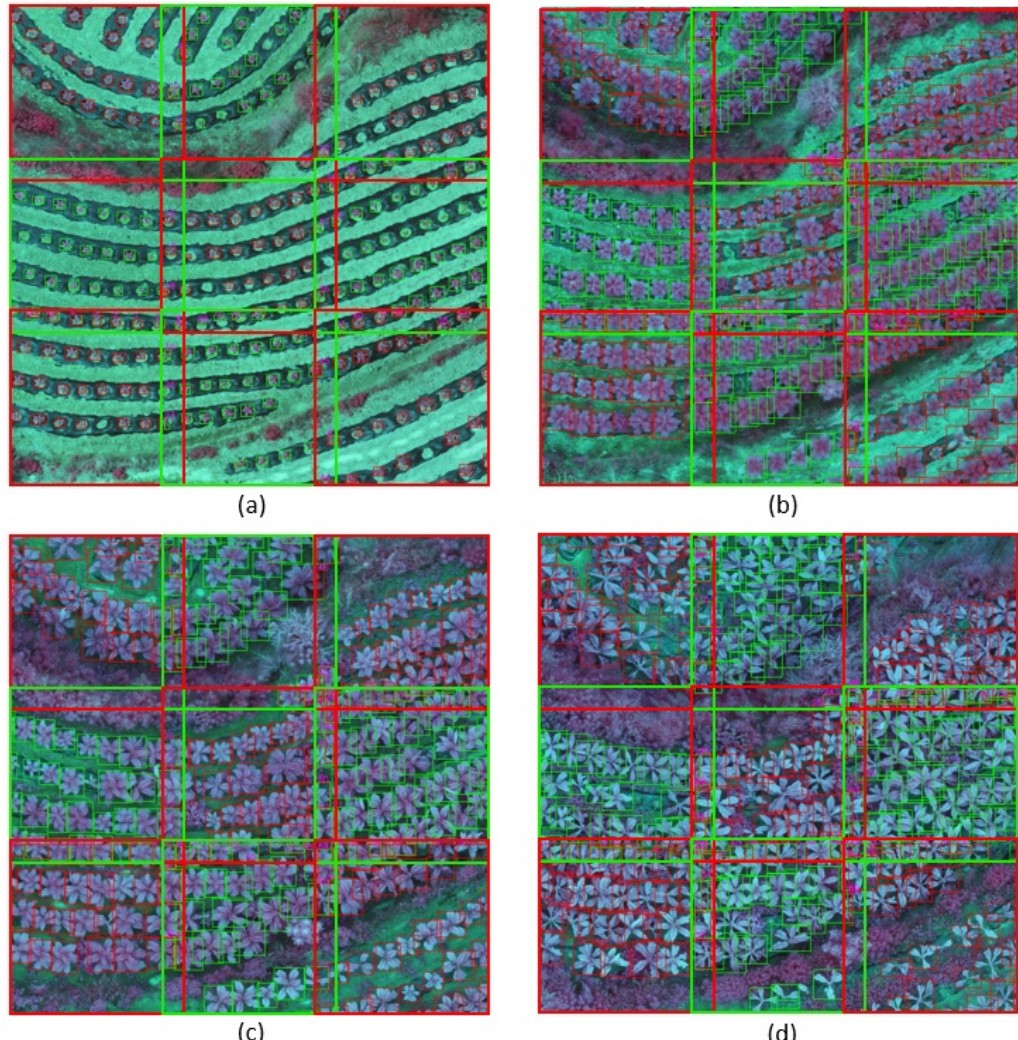

**Figure 11.** The plant detection and counting results of plot-s9. (**a**) is the result of 2 June 2023. (**b**) is the result of 14 July 2023. (**c**) is the result of 1 August 2023. (**d**) is the result of 23 August 2023. The red and green lines represent different slices, and we show a picture combination of nine slices in the experiment.

## 5. Discussion

1. So far, in the computer vision research field, the RGB format is the most commonly used, which includes three channels. For example, in OD, the main existing studies use RGB images. The multispectral data has been widely used in remote sensing and agriculture. However, it has not been fully used in DL in an agriculture-related area. Compared with the RGB data, the multispectral data contains more critical information concerning plants or vegetables. This means that more meaningful features for certain targets can be extracted by using different bands or band combinations. Some features are hidden at the back of the visible light bands but are very sensitive and have strong reflection in the invisible bands. The CNN has been verified as having a powerful ability to extract these features. Hence, it can also be used to extract features from multispectral images, which was performed here in this study.

2. PA is a wide topic. In addition to plant counting, many requirements can be conducted with the images, such as harvesting prediction, growth monitoring, healthy watching, etc. The analysis of nutrient content, moisture content, chlorophyll content, etc., have spacial indexes corresponding to certain bands, which can be executed with the UAV and multispectral data.

3.  The YOLOv8 is a mature OD neural network that has gained excellent performance. For using it in a specific application scenario, some adaption applications are needed. For example, in our work, firstly, the number of channels is modified depending on the input data. So, the architecture of YOLOv8 is also modified as well. In addition, the images in a specific area are not always kept to the standard size as required by the network. In our work, we combined and segmented the images' times to adapt the network and the task. Also, we conducted many experiments before and after the pre-processing and post-processing stage using YOLOv8.

While the study has demonstrated remarkable success in counting tobacco plants using the YOLOv8 network, it is crucial to acknowledge that the performance of deep learning-based methods, such as YOLOv8, can be influenced by the specific characteristics of the target plant species and the environmental conditions in which they are grown. Different plant species may have distinct shapes, sizes, and growth patterns that could affect the accuracy of object detection. Additionally, variations in environmental factors, such as lighting [50], soil types [51], and plant densities [52], could pose challenges to the model's general applicability [53]. Therefore, further research is needed to assess the adaptability of this method to a wider range of plant species and diverse environmental settings.

## 6. Conclusions

Plant counting is crucial monitoring process in agriculture for watching how many stands exist in a field. Accurate counting forms the foundation for yield estimation. In this work, we use YOLOv8 to count tobacco plants. Different from the existing works, we use the images captured via UAV that are multispectral data. By using transfer learning, only the small-scale data used in training gained excellent detection results. In order to make the model suitable for the life cycle of tobacco plants in the field, close to the application purpose, we conducted a large number of experiments and post-processed the output of YOLOv8. The results show that, by using multispectral data, the detection is more accurate than using traditional RGB images. The results of plant counting is really close to the ground truth. It indicates that using the UAV, multispectral data, and advanced DL technologies hold great promise in PA.

Our research aims to bridge the gap between cutting-edge technologies and practical applications in agriculture. By focusing on the life cycle of tobacco plants in the field, from germination to harvest, and tailoring our model accordingly, we strive to provide a comprehensive solution for farmers. The post-processing of YOLOv8 outputs further refines the accuracy of plant counting, bringing it in close alignment with ground truth.

The promising results obtained from our experiments underscore the significance of incorporating UAVs, multispectral data, and advanced deep learning technologies in precision agriculture. This integration holds tremendous potential not only for tobacco cultivation but also for a wide range of crops, paving the way for sustainable and efficient farming practices. As we continue to explore the synergies between technology and agriculture, the impact on yield estimation, resource optimization, and overall productivity is poised to be substantial.

**Author Contributions:** Conceptualization, H.L. and Z.Q.; methodology, H.L., Z.C. and Z.Q.; software, H.L., Z.C. and L.L.; validation, H.L., S.-K.T. and Z.C.; formal analysis, H.L. and S.-K.T.; investigation, L.L. and Z.Q.; resources, Z.Q.; data curation, Z.Q. and L.L.; writing—original draft preparation, H.L. and Z.C.; writing—review and editing, H.L., G.P. and Z.Q.; visualization, H.L.; supervision, H.L., G.P. and Z.Q.; project administration, H.L. and Z.Q.; funding acquisition, Z.Q. All authors have read and agreed to the published version of the manuscript.

**Funding:** This research was funded by the the Yunnan Fundamental Research Program of Agricultural spacial Projects (CN) (grant number 202301BD070001-008, 202101BD070001-053), the Natural Science Foundation of China (grant number 12163004) and the Fundamental Research Projects of Yunnan Provincial Department of Education (CN) (grant number 2022J0496).

**Institutional Review Board Statement:** Not applicable.

**Informed Consent Statement:** Not applicable.

**Data Availability Statement:** The data used in this research are collected and labeled by ourselves. We will publish the data after the article is accepted https://drive.google.com/drive/folders/1GpvSeP9E95GZxfhcp8w1FMuN2W50qY-0?usp=sharing (accessed on 16 November 2023). All codes and datasets in this study can be obtained on the website https://github.com/qzplucky/CountingTobaccoPlantsUAV (accessed on 16 November 2023) for easy access, analysis, and validation by researchers.

**Conflicts of Interest:** The authors declare no conflicts of interest.

## Abbreviations

The following abbreviations are used in this manuscript:

| | |
|---|---|
| PA | Precision Agriculture |
| UAV | Unmanned Aerial Vehicle |
| OD | Object Detection |
| YOLO | You only look once |
| CNN | Convolutional neural network |
| TP | True positive |
| FP | False Positive |
| FN | False Negative |
| TN | True Negative |
| IoU | Intersection over union |

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
