# Peer review of "Automated Counting of Tobacco Plants Using Multispectral UAV Data"

_agronomy, doi:10.3390/agronomy13122861_

Round 1

Reviewer 1 Report

Comments and Suggestions for Authors

Plant counting is a crucial field of study in precision agriculture (PA) and articles such as the one being reviewed hold significant worth. The authors effectively explain their research methodologies and highlight the reasons why it should capture the attention of a diverse range of readers.

However, there are few concerns:

1) Does the article mainly use images captured on a specific date, like 02/06/2023? It would have been more beneficial if the date used was after the planting date to depict the plant's age during the vegetative growth stage. Table 1 demonstrates this stage and could assist in estimating the count of tobacco plants, as shown in Table 2. Please refer to lines 361-363, as well as Tables 1 and 2.

2) The authors explicitly mention that their main focus is not to compare, but they do acknowledge that the best detection results are obtained when using a combination of red, green, and near-infrared bands. This finding suggests that utilizing specific band or band combinations can lead to superior results compared to traditional RGB images.

Specific problems and suggestions:

• Better, replace “Related Work” with “Review of Related Work”

• Better, replace “Methods” with “Research Methods”

• Line 125, replace “special” with “spatial”.

The comprehension of some tables and graphs can be improved through reworking. Additionally, the inclusion of the actual model being tested would be beneficial.

The author's commitment to extensively studying this subject is highly commendable, underscoring the significance of this research.

Comments on the Quality of English Language

It should be improved.

Author Response

Note: The revised part of the submitted revised paper has been shown in bold.

Reviewer 1:

Comments and Suggestions for Authors

Plant counting is a crucial field of study in precision agriculture (PA) and articles such as the one being reviewed hold significant worth. The authors effectively explain their research methodologies and highlight the reasons why it should capture the attention of a diverse range of readers.

However, there are few concerns:

  • Does the article mainly use images captured on a specific date, like 02/06/2023? It would have been more beneficial if the date used was after the planting date to depict the plant's age during the vegetative growth stage. Table 1 demonstrates this stage and could assist in estimating the count of tobacco plants, as shown in Table 2. Please refer to lines 361-363, as well as Tables 1 and 2.

Original paper:The e1 to e7 use the images taken on 02/06/2023. The e8 to e14 use the images taken on 29/06/2023. The e15 to e21 use the images taken on 14/07/2023. The e22 to e28 use the images taken on 01/08/2023. The e29 to e35 use the images taken on 23/08/2023...

Be changed to:The images from e1 to e7 were captured 30 days after the tobacco seedlings were transplanted into the field (on June 2, 2023). Images from e8 to e14 were taken 57 days after transplantation (on June 29, 2023). Images from e15 to e21 were captured 72 days after transplantation (on July 14, 2023). Images from e22 to e28 were taken 90 days after transplantation (on August 1, 2023). Images from e29 to e35 were captured 112 days after transplantation (on August 23, 2023). ...In the revised paper, lines 375-380.

2) The authors explicitly mention that their main focus is not to compare, but they do acknowledge that the best detection results are obtained when using a combination of red, green, and near-infrared bands. This finding suggests that utilizing specific band or band combinations can lead to superior results compared to traditional RGB images.

Specific problems and suggestions:

  • Better, replace “Related Work” with “Review of Related Work”

The “Related Work”  in the original paper has been revised to “Review of Related Work”   (modified in line 126).

  • Better, replace “Methods” with “Research Methods”

“Methods”  in the original paper have been revised to “Research Methods” (modified in line 207).

  • Line 125, replace “special” with “spatial”.

Original paper:In recent years, the UAV is more frequently used for plant counting because the captured data has higher special and temporal resolution [20].

special in the original paper has been revised to spatial (129 lines).

Be changed to: In recent years, the UAV is more frequently used for plant counting because the captured data has higher spatial and temporal resolution [23].

The comprehension of some tables and graphs can be improved through reworking. Additionally, the inclusion of the actual model being tested would be beneficial.

The author's commitment to extensively studying this subject is highly commendable, underscoring the significance of this research.

Reviewer 2 Report

Comments and Suggestions for Authors

The reviewed work is an in-depth study, including the use of multispectral images for counting plants; the author's multispectral data set on the growth of tobacco plants; as well as a modified architecture of the modern YOLOv8 neural network adapted to input data of different color ranges and a post-processing algorithm for counting plants on stretched fields. The absolute advantage of the study is the willingness of the authors to present the collected dataset and the source codes of the modified YOLOv8 neural network.

Remarks:

1. It is necessary to explain the structure of the choice of parameters of numerous experiments given in the article.

2. In section 6. Discussion, the interesting results obtained should be compared with the materials of other authors.

3. The article states that the results obtained can be used to predict the crop yield, but this requires clarification, taking into account the need for quantitative assessment of the yield level.

4. The methodology and the results obtained in the study of tobacco plants are of considerable interest for the study of this agricultural culture. However, the authors' opinion is of interest regarding the possibility of using the described technique in the process of growing other crops, which is desirable to be discussed in section 7.

Author Response

Note: The revised part of the submitted revised paper has been shown in bold.

Reviewer 2:

Comments and Suggestions for Authors

The reviewed work is an in-depth study, including the use of multispectral images for counting plants; the author's multispectral data set on the growth of tobacco plants; as well as a modified architecture of the modern YOLOv8 neural network adapted to input data of different color ranges and a post-processing algorithm for counting plants on stretched fields. The absolute advantage of the study is the willingness of the authors to present the collected dataset and the source codes of the modified YOLOv8 neural network.

Remarks:

1. It is necessary to explain the structure of the choice of parameters of numerous experiments given in the article.

Original paper:In experiments, we conduct 103 experiments. These experiments can be approximately summarized into three categories: the OD performance, the accuracy of plant counting of slices, the accuracy of plant counting of a big plot.

In the revised paper, we have supplemented the Settings of relevant parameters in the experiment in detail. In the revised paper, lines 326-331:

Be changed to: In experiments, we conduct 103 experiments. We adopted the relevant settings of the basic network (YOLOv8), including the settings of basic parameters. Through a series of experiments and comparisons, we found that when the batch of the original basic network is set to 128 and epoch is set to 200, the model performs best and has smaller fluctuations. In view of this, we adjusted batch and epoch to better fit our multispectral dataset while keeping other parameters of the base network unchanged.These experiments can be approximately summarized into three categories: the OD performance, the accuracy of plant counting of slices, the accuracy of plant counting of a big plot.

 2. In section 6. Discussion, the interesting results obtained should be compared with the materials of other authors.

Note: In this study, we used UAV-based multi-spectral smoke plant numbers for statistical analysis, focusing on images of different channels. According to the latest research data, YOLOv8 achieved the optimal accuracy in the direction of target detection, so we extended the use of YOLOv8 network. Because the accuracy of the basic network was already optimal, we did not carry out precision comparison experiments with other authors' networks.

 3. The article states that the results obtained can be used to predict the crop yield, but this requires clarification, taking into account the need for quantitative assessment of the yield level.

Original paper:It can be used for predicting the yields and the efficiency of cultivation...

We have revised the relevant statement of the original paper in line 520 of the revised paper.

Be changed to:Accurate counting forms the foundation for yield estimation...

 4. The methodology and the results obtained in the study of tobacco plants are of considerable interest for the study of this agricultural culture. However, the authors' opinion is of interest regarding the possibility of using the described technique in the process of growing other crops, which is desirable to be discussed in section 7.

Reviewer 3 Report

Comments and Suggestions for Authors

Dear Authors, thank you for providing materials about plant counting using YOLOv8. I found this study very interesting. There are some comments below:

The introduction seems to be difficult to understand. The flow is controversial, the paragraph 107-114 looks like a continuation or repetition of the introduction's beginning. Two paragraphs about tobacco plants should be put after line 114. So, everything about PA and crop counting will be together without interruption for tobacco explanation. Please check the introduction again and consider revising it.

27: Yield estimation should start from 1)?

26–50: I believe there should be more references.

74: "In the existing works" references must be applied here.

2. Related work: This section provided a strong literature review.

3. Methods: It is an explanation of how the network works. Are there any references or sources?

7. Conclusion: In my opinion, the conclusion is very limited.

The overall materials were presented very well. However, there are many mistakes and flaws in the text and captions. I believe those issues will be solved in the next step.

Thanks, authors, for the provided research. I wish good luck on this topic, it's very interesting and valuable for future agriculture.

Author Response

Note: The revised part of the submitted revised paper has been shown in bold.

Comments and Suggestions for Authors

Dear Authors, thank you for providing materials about plant counting using YOLOv8. I found this study very interesting. There are some comments below:

The introduction seems to be difficult to understand. The flow is controversial, the paragraph 107-114 looks like a continuation or repetition of the introduction's beginning. Two paragraphs about tobacco plants should be put after line 114. So, everything about PA and crop counting will be together without interruption for tobacco explanation. Please check the introduction again and consider revising it. 

Original paper:Tobacco is a kind of highly valuable crop due to its economic significance in the global market. Tobacco cultivation and the tobacco industry contribute significantly to the economies of many countries. It provides income and employment opportunities for farmers, laborers, and workers involved in various stages of production, processing, and distribution. The global tobacco market is substantial means that tobacco products continue to be in demand worldwide. In addition, the tobacco is an important commodity for export earnings and value-added products [16].

The cultivation of tobacco occurs annually. The plant is germinated in cold frames or hotbeds firstly and then transplanted to the field. The time from transplanting to harvest depends on the breeds, generally around 60 to 90 days, but in the range of 45 to 120 days [16]. During the cultivation in the field, the UAVs can be used for monitoring the number of stands after the transplanting, as well as the number of plants during the growth in field. The yield of tobacco largely depends on the number of viable tobacco plants because the leaves are the harvest of tobacco plant. Hence, the tobacco plant counting is also relevant for the yield estimation.

In recent years, along with the blooming of the advanced technologies, such as artificial intelligence, computer vision, machine learning, deep learning etc. they have been greatly introduced into agriculture industry [17]. Many traditional tasks in agriculture, such like plant counting, plant disease identification ect. have achieved at excellent performance [18,19]. Deep learning is a kind of data-driven method which generally requires large-scale data to train the network. By using UAV, the data collection work is easier and faster. Combined with computer vision, the automated plant counting is promising by using deep learning method.

  The contents of lines 107-114 of the original paper have been moved to lines 95-102 of the revised paper before the corresponding lines 92 of the original paper. Modified as follows:

Be changed to :In recent years, along with the blooming of the advanced technologies, such as artificial intelligence, computer vision, machine learning, deep learning etc. they have been greatly introduced into agriculture industry [19]. Many traditional tasks in agriculture, such like plant counting, plant disease identification ect. have achieved at excellent performance [20,21]. Deep learning is a kind of data-driven method which generally requires large-scale data to train the network. By using UAV, the data collection work is easier and faster. Combined with computer vision, the automated plant counting is promising by using deep learning method.}

Tobacco is a kind of highly valuable crop due to its economic significance in the global market. Tobacco cultivation and the tobacco industry contribute significantly to the economies of many countries. It provides income and employment opportunities for farmers, laborers, and workers involved in various stages of production, processing, and distribution. The global tobacco market is substantial means that tobacco products continue to be in demand worldwide. In addition, the tobacco is an important commodity for export earnings and value-added products [22].

The cultivation of tobacco occurs annually. The plant is germinated in cold frames or hotbeds firstly and then transplanted to the field. The time from transplanting to harvest depends on the breeds, generally around 60 to 90 days, but in the range of 45 to 120 days [22]. During the cultivation in the field, the UAVs can be used for monitoring the number of stands after the transplanting, as well as the number of plants during the growth in field. The yield of tobacco largely depends on the number of viable tobacco plants because the leaves are the harvest of tobacco plant. Hence, the tobacco plant counting is also relevant for the yield estimation.

27: Yield estimation should start from 1)?   

Original paper:Yield estimation:Accurate plant counting helps estimate crop yields, which is crucial for agricultural planning, resource allocation, and market forecasting. It enables farmers to make informed decisions regarding harvesting, storage, and marketing of their produce.1)...

We have revised the relevant statement of the original paper in line 30 of the revised paper.

Be changed to :Accurate plant counting helps estimate crop yields, which is crucial for agricultural planning, resource allocation, and market forecasting. It enables farmers to make informed decisions regarding harvesting, storage, and marketing of their produce. Yield estimation should start from 1)...

26–50: I believe there should be more references.  

Original paper:

  • Crop management: Plant counting provides essential information for effective crop management. By knowing the population density of plants, farmers can optimize irrigation, fertilization, and pest control practices tailored to specific crop requirements. It helps ensure that plants receive adequate resources, leading to improved growth, health, and yield....

4) Disease and pest monitoring: Plant counting can assist in early detection and monitoring of plant diseases and pests. By regularly counting plants, farmers or researchers can identify and track the spread of diseases or infestations. Timely intervention measures can be implemented to prevent further damage and minimize crop losses.

Relevant references have been added in lines 26-50 as strong support, respectively in lines 33 and 46 of the revised paper.

Be changed to :

33 lines:1)Crop management: Plant counting provides essential information for effective crop management. Knowing before harvesting how many plants have emerged and how they are growing is key in optimizing labour and efficient use of resources [5]. By knowing the population density of plants, farmers can optimize irrigation, fertilization, and pest control practices tailored to specific crop requirements.

46 lines:4) Disease and pest monitoring: Plant counting can assist in early detection and monitoring of plant diseases and pests.Plant counting has a wide range of applications in agriculture, such as crop management, yield estimation, disease and pest monitoring, etc [6].

74: "In the existing works" references must be applied here.  

Original paper: In the existing works of plant counting, the most frequently used image is the visible whit three visible (VIS) bands:..

We have revised the relevant statement of the original paper in line 77 of the revised paper.

A reference has been added, modified to : In the existing works [5,15] of plant counting, the most frequently used image is the visible whit three visible (VIS) bands:..

  1. Related work: This section provided a strong literature review.
  2. Methods: It is an explanation of how the network works. Are there any references or sources?

Original paper :

The framework of YOLOv8 is as shown in Figure 2[44]. YOLOv8 is a deep learning-based OD algorithm that enables fast, accurate, and robust OD and instance segmentation on high-resolution images. It includes three components: backbone part, neck part and head part.

The backbone network is a convolutional neural network used for extracting image features, and YOLOv8 employs a C2f structure, which enhances gradient flow and feature fusion, thereby improving feature representation capabilities.

After obtaining the bounding box coordinates, YOLOv8 uses a new positive and negative sample matching strategy, which can determine positive and negative samples based on the intersection over union (IoU) between the bounding box and the real label.

Relevant references have been added to the above three sentences in the method part as a strong support, and relevant references have been introduced in lines 209, 217 and 269 of the revised paper respectively.

Be changed to :

209 lines:The YOLO algorithms series has become a widely used algorithm as a one-step algorithm for objects detection [47].In this study, we used YOLOV8 as the baseline network.

The framework of YOLOv8 is as shown in Figure 2[48]. YOLOv8 is a deep learning-based OD algorithm that enables fast, accurate, and robust OD and instance segmentation on high-resolution images. It includes three components: backbone part, neck part and head part.

217 lines:The backbone network is a convolutional neural network used for extracting image features. YOLOv8 uses a similar backbone as YOLOv5 with some changes on the CSPLayer, now called the C2f module [44], which enhances gradient flow and feature fusion, thereby improving feature representation capabilities.

269 lines:After obtaining the bounding box coordinates, YOLOv8 uses the loss function of CIoU [49] and DFL [50] for boundary box loss, and binary cross entropy for classification loss. It is a new positive and negative sample matching strategy, which can determine positive and negative samples according to the IoU (intersection over union) between the boundary box and the real label.

  1. Conclusion: In my opinion, the conclusion is very limited.

Original paper :

Plant counting is crucial monitoring in agriculture for watching how many stands are existing in the field. Accurate counting forms the foundation for yield estimation. In this work, we use the YOLOv8 to count tobacco plants. Different from the existing works, we use the images captured by UAV that are multispetral data. By using transfer learning, only small-scale data used in training has gained excellent detection results. In order to make the model suitable for the life cycle of tobacco plants in the field, close to the application purpose, we conducted a large number of experiments and post-processed the output of YOLOv8. The results show that by using multispectral data, the detection is more accurate than using the traditional RGB images. The results of plant counting is really close to the ground truth. It indicates that using the UAV, multispectral data and advanced DL technologies holds great promise in PA.

Our conclusion has been expanded. In the revised paper, lines 530-541 are our new extension, which is revised as follows:

Be changed to :

Plant counting is crucial monitoring in agriculture for watching how many stands are existing in the field. Accurate counting forms the foundation for yield estimation. In this work, we use the YOLOv8 to count tobacco plants. Different from the existing works, we use the images captured by UAV that are multispetral data. By using transfer learning, only small-scale data used in training has gained excellent detection results.In order to make the model suitable for the life cycle of tobacco plants in the field, close to the application purpose, we conducted a large number of experiments and post-processed the output of YOLOv8. The results show that by using multispectral data, the detection is more accurate than using the traditional RGB images. The results of plant counting is really close to the ground truth. It indicates that using the UAV, multispectral data and advanced DL technologies holds great promise in PA.

Our research aims to bridge the gap between cutting-edge technologies and practical applications in agriculture. By focusing on the life cycle of tobacco plants in the field, from germination to harvest, and tailoring our model accordingly, we strive to provide a comprehensive solution for farmers. Post-processing of YOLOv8 outputs further refines the accuracy of plant counting, bringing it in close alignment with ground truth.

The promising results obtained from our experiments underscore the significance of incorporating UAVs, multispectral data, and advanced deep learning technologies in precision agriculture. This integration holds tremendous potential not only for tobacco cultivation but also for a wide range of crops, paving the way for sustainable and efficient farming practices. As we continue to explore the synergies between technology and agriculture, the impact on yield estimation, resource optimization, and overall productivity is poised to be substantial.

The overall materials were presented very well. However, there are many mistakes and flaws in the text and captions. I believe those issues will be solved in the next step.

Thanks, authors, for the provided research. I wish good luck on this topic, it's very interesting and valuable for future agriculture.

Reviewer 4 Report

Comments and Suggestions for Authors

The manuscript titled "Automated Counting of Tobacco Plants Using Multispectral UAV Data" addresses the significant issue of plant counting in precision agriculture using unmanned aerial vehicles (UAVs) equipped with multispectral sensors. The authors employ deep learning object detection (OD) techniques, specifically adapting the YOLOv8 network, to automatically count tobacco plants in the field using multispectral images. The study investigates various band combinations and demonstrates the effectiveness of the Red+Green+NIR combination in achieving the best detection results, surpassing traditional RGB images. The paper also introduces an algorithm for handling whole plot images, achieving a remarkable counting accuracy of 99.53%. The availability of the codes and datasets for researchers on the provided website enhances the reproducibility and accessibility of the study.

Overall, the manuscript is well-structured and aligns with the scope of Agronomy, addressing a contemporary issue in precision agriculture. The use of UAVs and multispectral data for plant counting is of great significance, and the application of deep learning techniques adds a valuable dimension to this research. The adaptation of the YOLOv8 network to accommodate different band combinations demonstrates a thorough approach to method development.

1.      Main Question:

The main question addressed by the research is the development and evaluation of an automated plant counting method using multispectral Unmanned Aerial Vehicle (UAV) data, specifically for counting tobacco plants. The study aims to assess the performance of deep learning-based object detection in the context of precision agriculture and the use of multispectral imagery.

2.      Originality and Relevance:

The topic of automated plant counting using UAV data is highly relevant and valuable in the field of precision agriculture. This paper addresses a specific gap by focusing on tobacco plant counting, which has distinct requirements due to its unique growth characteristics. The utilization of multispectral sensors for this purpose is relatively novel and adds to the originality of the research.

3.      Contribution:

The reserch makes a significant contribution to the subject area by demonstrating the feasibility and effectiveness of using multispectral UAV data and deep learning for plant counting. The modification of the YOLOv8 network to accommodate different band combinations is a valuable methodological contribution. Additionally, the comparison of Red+Green+NIR band combination to traditional RGB imagery highlights the superiority of specific band combinations in obtaining better results.

4.      Methodology and Improvements:

The methodology employed in this study is robust and well-described. However, a few specific improvements can be considered:

a. It would be beneficial to provide more details on the extensive data pre-processing work, as this is a critical component of the method's success.

b. The authors should discuss the limitations of their approach, including any potential challenges or failure cases in plant counting.

c. Including a section on the computational requirements and runtime of the method would be helpful for readers assessing its practicality.

One of the strengths of this work is the comprehensive presentation of the methodology, which includes details about the deep learning network architecture, band combinations, and data preprocessing. The results are promising, especially in the context of detecting tobacco plants in the field, and the high counting accuracy is commendable.

5.      Results and Discussion

However, there are areas where the manuscript could be improved. Firstly, the abstract lacks a clear statement of the study's significance and potential contributions to precision agriculture, which is essential to grab the reader's attention. Additionally, while the authors discuss the superiority of the Red+Green+NIR band combination, a more in-depth analysis of why this combination is more effective than RGB or other alternatives would enhance the paper's scientific rigor. Furthermore, the study could benefit from a discussion of potential limitations and challenges, such as the generalizability of the method to other plant species or varying environmental conditions.

For example:

Line 490: While the study has demonstrated remarkable success in counting tobacco plants using the YOLOv8 network, it is crucial to acknowledge that the performance of deep learning-based methods, such as YOLOv8, can be influenced by the specific characteristics of the target plant species and the environmental conditions in which they are grown. Different plant species may have distinct shapes, sizes, and growth patterns that could affect the accuracy of object detection. Additionally, variations in environmental factors, such as lighting [46], soil types [47], and plant densities [48], could pose challenges to the model's general applicability [49, 50]. Therefore, further research is needed to assess the adaptability of this method to a wider range of plant species and diverse environmental settings.

46. Rodríguez-Yzquierdo, G.; Olivares, B.O.; Silva-Escobar, O.; González-Ulloa, A.; Soto-Suarez, M.; Betancourt-Vásquez, M. Mapping of the Susceptibility of Colombian Musaceae Lands to a Deadly Disease: Fusarium oxysporum f. sp. cubense Tropical Race 4. Horticulturae 2023, 9, 757. https://doi.org/10.3390/horticulturae9070757

47. Orlando, O.; Rey, J.C.; Lobo, D.; Navas-Cortés, J.A.; Gómez, J.A.; Landa, B.B. Fusarium Wilt of Bananas: A Review of Agro-Environmental Factors in the Venezuelan Production System Affecting Its Development. Agronomy 2021, 11, 986. https://doi.org/10.3390/agronomy11050986

48. Vega, A.; Rueda Calderón, M.A.; Montenegro-Gracia, E.; Campos, O.; Araya-Almán, M.; Marys, E. 2022. Prediction of Banana Production Using Epidemiological Parameters of Black Sigatoka: An Application with Random Forest. Sustainability 14, 14123. https://doi.org/10.3390/su142114123

59. Campos, O. Evaluation of the Incidence of Banana Wilt and its Relationship with Soil Properties. In: Banana Production in Venezuela. The Latin American Studies Book Series. 2023; Springer, Cham. https://doi.org/10.1007/978-3-031-34475-6_4

6.      Consistency of Conclusions:

The conclusions drawn in the paper are generally consistent with the evidence and arguments presented. The high counting accuracy of 99.53% is impressive and supports the potential of this method for precision agriculture. However, the authors should emphasize the limitations and potential sources of error to provide a more balanced view of the results.

7.      References:

The references appear to be appropriate and relevant to the research topic. The citation of YOLOv8 and the use of multispectral data is well-supported.

Author Response

Note: The revised part of the submitted revised paper has been shown in bold.

Line 490: While the study has demonstrated remarkable success in counting tobacco plants using the YOLOv8 network, it is crucial to acknowledge that the performance of deep learning-based methods, such as YOLOv8, can be influenced by the specific characteristics of the target plant species and the environmental conditions in which they are grown. Different plant species may have distinct shapes, sizes, and growth patterns that could affect the accuracy of object detection. Additionally, variations in environmental factors, such as lighting [46], soil types [47], and plant densities [48], could pose challenges to the model's general applicability [49, 50]. Therefore, further research is needed to assess the adaptability of this method to a wider range of plant species and diverse environmental settings.   

This statement has been added in the paper, in the revised paper lines 508-517.

While the study has demonstrated remarkable success in counting tobacco plants using the YOLOv8 network, it is crucial to acknowledge that the performance of deep learning-based methods, such as YOLOv8, can be influenced by the specific characteristics of the target plant species and the environmental conditions in which they are grown. Different plant species may have distinct shapes, sizes, and growth patterns that could affect the accuracy of object detection. Additionally, variations in environmental factors, such as lighting [52], soil types [53], and plant densities [54], could pose challenges to the model's general applicability [55]. Therefore, further research is needed to assess the adaptability of this method to a wider range of plant species and diverse environmental settings.